# Evolving Graphical Planner: Contextual Global Planning for Vision-and-Language Navigation

**Zhiwei Deng    Karthik Narasimhan    Olga Russakovsky**
Department of Computer Science
Princeton University
{zhiweid, karthikn, olgarus}@cs.princeton.edu

## Abstract

The ability to perform effective planning is crucial for building an instruction-following agent. When navigating through a new environment, an agent is challenged with (1) connecting the natural language instructions with its progressively growing knowledge of the world; and (2) performing long-range planning and decision making in the form of effective exploration and error correction. Current methods are still limited on both fronts despite extensive efforts. In this paper, we introduce the *Evolving Graphical Planner (EGP)*, a model that performs global planning for navigation based on raw sensory input. The model dynamically constructs a graphical representation, generalizes the action space to allow for more flexible decision making, and performs efficient planning on a proxy graph representation. We evaluate our model on a challenging Vision-and-Language Navigation (VLN) task with photorealistic images, and achieve superior performance compared to previous navigation architectures. For instance, we achieve a 53% success rate on the test split of the Room-to-Room navigation task [1] through pure imitation learning, outperforming previous navigation architectures by up to 5%.

## 1   Introduction

Recent work has made remarkable progress towards building autonomous agents that navigate by following instructions [2, 3, 4, 5, 6, 7, 8, 9, 10] and constructing memory structures for maps [11, 12, 13]. An important problem setting within this space is the paradigm of *online navigation*, where an agent needs to perform navigation based on goal descriptions in an unseen environment using a limited number of steps [14, 1].

In order to successfully navigate through an *unseen* environment, an agent needs to overcome two key challenges. First, the instructions given to the agent are natural language descriptions of the goal and the landmarks along the way; these descriptions need to be grounded onto the evolving visual world that the agent is observing. Second, the agent needs to perform non-trivial planning over a large action space, including: 1) deciding which step to take next to resolve ambiguities in the instructions through novel observations and 2) gaining a better understanding of the environment layout in order to progress towards the goal or recover from its prior mistakes. Notably this planning requires not only selecting from an increasingly large set of possible actions but also performing complex long-term reasoning.

Existing work tackles only one or two components of the above and may require additional pre-processing steps. Some are constrained to use local control policies [14, 2] or use rule-based algorithms such as beam or $A^*$ search [2, 15, 14] to perform localized path corrections. Others focus on processing long-range observations instead of actions [16] or employ offline pre-training schemes to learn topological structures [12, 13, 17]. This is challenging since accurate construction of graphs is non-trivial and requires special adaptation to work during real-time navigation [13, 17].

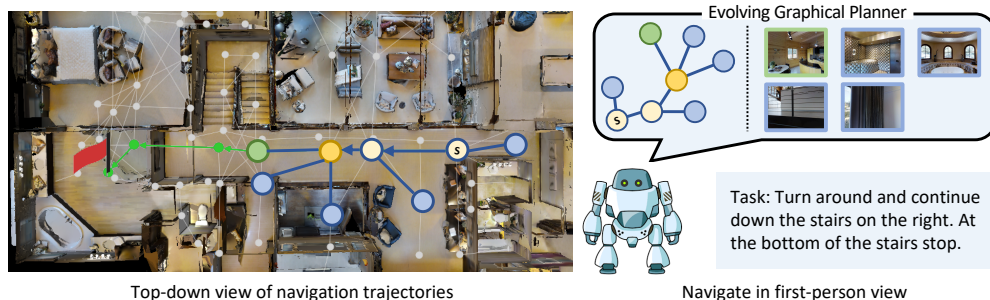

Top-down view of navigation trajectories          Navigate in first-person view

Figure 1: Under the guidance of natural language instruction, the autonomous agent needs to navigate through the environment from the start state to the target location (red flag). Our proposed Evolving Graphical Planner (EGP) constructs a dynamic representation and makes decisions in a global action space (right). With the EGP, the agent, currently in the orange node, maintains and reasons over the evolving graph to select the next node to visit (green) from possible choices (blue).

In this paper, we propose the *Evolving Graphical Planner (EGP)* (Figure 1), which 1) dynamically constructs a graphical map of the environment in an online fashion during exploration and 2) incorporates a global planning module for selecting actions. *EGP* can operate directly on raw sensory inputs in partially observable settings by building a structured representation of the geometric layout of the environment using discrete symbols to represent visited states and unexplored actions. This expressive representation allows our agent to choose from a greater number of global actions conditioned on the text instruction and perform course corrections if needed.

Incorporating a global navigation module is challenging since we do not always have access to ground truth supervisions from the environment. Further, the ever-expanding size of the global graphs requires a scalable action selection module. To solve the first challenge, we introduce a novel method for training our planning modules using imitation learning – this allows the agent to efficiently learn how to select global actions and backtrack when necessary. For the second, we introduce *proxy graphs*, which are local approximations of the entire map and allow for more scalable planning. Our entire model is end-to-end differentiable under a pure imitation learning framework.

We test *EGP* on two benchmarks for 3D navigation with instructions – Room-to-Room [1] and Room-for-Room [18]. Our model outperforms several state-of-the-art *backbone architectures* on both datasets – e.g., on Room-to-Room, we achieve a 5% improvement in Success Rate over the Regretful agent [19]. We also perform a series of ablation studies on our model to justify the design choices. Our implementation is available at `https://github.com/Lucas2012/EvolvingGraphicalPlanner`.

## 2   Related work

**Embodied Navigation Agent** Many recent papers develop neural architectures for navigation tasks [19, 2, 14, 20, 21, 22, 16, 23, 24, 25]. Vision-and-Language Navigation (VLN) [1, 5] is one representative task that focuses on language-driven navigation across photo-realistic 3D environments. Anderson et al. [1] propose the *Room-to-Room* benchmark and an attention-based sequence-to-sequence method. Fried et al. [2] extend the model by a pragmatic agent with the ability to synthesize data through a speaker model. With an emphasize on grounding, the self-monitoring agent [14] adopts a co-grounding module and a progress estimation auxiliary task for more progress-sensitive alignment. Similarly, an intrinsic reward [26] is introduced to improve cross-modal alignment for navigation agent. Ma et al. extends the existing works towards a graph search like algorithm by adding one-step regretful action. Anderson et al. [27] proposes an agent formulated under Bayesian filtering with a global mapper. Hand-crafted decoding algorithms are also used as a post-processing technique but may lead to long trajectories and difficulty on joint optimization [2, 15, 28].

**Navigation memory structures** A recent emerging trend of navigation focuses on extending the agents with different types of memory structures. *Simple structures:* Parisotto et al. [11] propose a

tensor-based memory structure with operations that agent can use to access and perform navigation. Fang et al. [16] adopts transformer to extract historical memory information stored in a sequence. *Topological structures:* The landmark-based topological representation has shown to be effective in pre-exploration based navigation tasks without the need of externally provided camera poses or ego-motion information[12]. Laskin et al. proposes methods to sparsify the graphical memory through consistency checkings and graph cleanups [13]. Liu et al. uses a contrastive energy model for building more accurate edges in the memory graph [17].

**Graphical representation** Graph-based methods have been shown to be an effective intermediate representation for information exchange [29, 30, 31, 32, 33]. In image and video understanding, graphical representation is used for visual question answering [34, 35], video captioning [36] or action recognition[37, 38]. In robotics, Huang et al. [39] propose Neural Task Graph (NTG) as an intermediate modularized representation leading to better generalization. Graph Neural Networks are demonstrated to be effective in learning structured policies [26] and automatic robot design [40].

**Supervision strategies for imitation learning** The proper training of sequential models is challenging due to the drifting issues [41]. DAgger [41] proposes to aggregate datasets with expert supervision provided for the sequences samples from student models. Scheduled sampling [42] tackles this problem through mixing the samples from both ground truth and student models. Professor forcing [43] shows a more effective approach through adversarial domain adaptation. OCD [44] adopts the online computed characters as ground truth for speech recognition. In this paper, we instead propose a graph-augmented strategy to provide expert supervisions, which alleviates the mismatch issue between instructions and new-computed expert trajectories [14].

## 3  Model

**Problem definition and setup** We follow the standard instruction-following navigation problem setting [1]. Given the demonstration dataset $\mathcal{D} = \{(\boldsymbol{x}_i, \boldsymbol{\tau}_i^*, \text{ENV}_i)\}_{i=1}^{|\mathcal{D}|}$, where $\boldsymbol{x}_i$ is the language instruction with length $|\boldsymbol{x}_i|$, $\boldsymbol{\tau}^*$ is the expert navigation trajectories $(a_1^*, a_2^*, ..., a_{|\boldsymbol{\tau}^*|}^*)$ and $\text{ENV}_i$ is the environment paired with the data, the agent is trained to imitate the expert behaviours to correctly follow the instruction and navigate to the goal location. At each navigation step $t$, the agent is located at the state $\boldsymbol{s}_t$ with observations $\boldsymbol{o}_t$ from the environment and performs an action $a_t \in \mathcal{A}_{\boldsymbol{s}_t}$, where $\mathcal{A}_{\boldsymbol{s}_t}$ is the decision space at state $\boldsymbol{s}_t$. In our task, decision space is the set of navigable locations [2].

To set up an agent, we build upon a basic navigation architecture in [14] and utilize the language encoder and the attention-based decoder. The agent encodes the language instruction $\boldsymbol{x}$ into a hidden encoding $\boldsymbol{c} = (\boldsymbol{c}_1, \boldsymbol{c}_2, ..., \boldsymbol{c}_{|\boldsymbol{x}|})$ through an LSTM [45]. Conditioned on $\boldsymbol{c}$, the agent uses an attention-based decoder to model the distribution over the trajectory $p(\boldsymbol{\tau}|\boldsymbol{x}, \text{ENV})$. At every step, the decoder takes in the encoding $\boldsymbol{c}$, the observation $\boldsymbol{o}_t$ and a maintained hidden memory $\boldsymbol{h}_{t-1}$ to produce the per-step action probability distribution $p(a_t|\boldsymbol{h}_{t-1}, \boldsymbol{o}_t, \boldsymbol{c})$. Note that such an navigation agent suffers from the constrained local action set and lacks the ability to perform long-range planning over the navigation space and to effectively correct errors along the navigation.

**Our approach** In this section, we introduce our *Evolving Graphical Planner (EGP)*, an end-to-end global planner that navigates a agent through a new environment via re-defining the decision space and performing long-term plannings over graphs. With the graphical representation, the agent accumulates the knowledge about the unseen environment and has access to the entire action space. Each long-distance navigation is then reduced to one-step decision making, leading to easier and more efficient exploration and error correction. We also show that the graphical representation elicits a new supervision strategy for effective training of the imitation agent, which alleviates the mismatch issue in standard navigation agent training [14]. The global planning is performed on a efficient proxy representation, making the model scalable to navigation with longer horizon.

The proposed model consists of two core components: 1) an evolving graphical representation that generalizes the local action space; and 2) a planning core that performs multi-channel information propagation on a proxy representation. Starting from the initial information, the EGP agent gradually expands an internal representation of the explored space (Section 3.1) and perform planning efficiently over the graphs (Section 3.2).

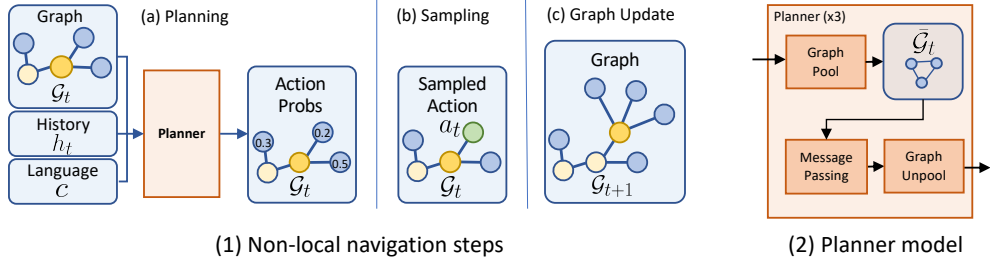

(1) Non-local navigation steps            (2) Planner model

Figure 2: Overall scheme of the Evolving Graphical Planner model. (1): Our graphical representation progressively expands during navigation (light-yellow = visited nodes; orange = current node; blue = potential action nodes; green = selected node). Based on the graphical representation, the agent performs planning over the actions, and selects the next action through student sampling. The top-K nodes on the current state will be kept in the graph. Note that the STOP action is also affiliated to the internal node as a leaf node, enabling the agent to travel back and stop at a previous visited location. (2): The model performs multi-channel planning on a proxy graph pooled from the entire graph representation. The refined node states are unpooled back to the entire graph and used to compute the probability distribution over actions. Best viewed in color.

## 3.1 Evolving Graphical Planner: Graphical representation

We start with introducing the notations of the representation. Let $\mathcal{G}_t = \{\mathcal{V}_t, \mathcal{E}_t, \boldsymbol{M}_t\}$ denote a graph at time $t$, where $\mathcal{V}_t = \{\boldsymbol{v}^1, \boldsymbol{v}^2, ..., \boldsymbol{v}^{|\mathcal{V}_t|}\}, \boldsymbol{v}^i \in \mathbb{R}^d$ is the set of node embeddings, $\mathcal{E}_t = \{\boldsymbol{e}^{ij}\}, \boldsymbol{e}^{ij} \in \mathbb{R}^d$ is the set of edge embeddings between node $i$ and node $j$, and $\boldsymbol{M}_t \in \mathbb{R}^{|\mathcal{V}_t| \times |\mathcal{V}_t| \times |\mathcal{F}|}$ is the graph connectivity tensor with function types. In our graph, nodes are separated into two types: the leaf nodes represent the possible actions (e.g. navigable locations) and internal nodes represent the visited locations, as shown in figure 2.

**Graph construction** The agent builds up the graphical representation progressively during navigation. Initialized as the empty set, the graph $\mathcal{G}_t$ expands the node set, edge set and connectivity function tensor through the observations and local connection information given by the environment. When receiving an observation $\boldsymbol{o}_t$ and the set of information $\{\boldsymbol{o}_{a_t}\}_{\boldsymbol{s}_t}$ over possible actions $\{a_t\}_{\boldsymbol{s}_t}$ at the new state $\boldsymbol{s}_t$, the agent maps the current location information $\boldsymbol{o}_t$ and action information $\boldsymbol{o}_{a_t}$ through two separate neural networks, resulting to the node embedding $\boldsymbol{v}_{\boldsymbol{s}_t}$ and $\{\boldsymbol{v}_{a_k}\}_{\boldsymbol{s}_t}$. The incoming and outgoing edges of new nodes are determined through the local map information and the moving direction of the agent. There are three function types considered and stored in the tensor $\boldsymbol{M}_t$: the forward and backward directions between two visited nodes, and the connection from the visited node to the leaf node. With the new graph $\mathcal{G}_{t+1}$, the agent continues navigation and the loop repeats. To reduce the memory cost, the model has the option to selectively add nodes to the graph. We use top-K leaf nodes, ranked by confidence scores in policy distribution, to expand the graph, as shown in fig. 2.

**Navigation with graph jump** With graph $\mathcal{G}_t$ representing the entire action space, the agent can easily choose to navigate to execute actions that have not been explored in previous visited locations. For the proposed action $a_t$ from the planner, the agent computes the shortest-path route based on the graph $\mathcal{G}_t$ and plans the navigation route $\boldsymbol{\tau}'$. This allows the agent to "jump" around the full graph-defined action space and execute the unexplored actions through a single-step decision. The long-range decision space also makes error-correction easier for the agent: a single step decision is all the agent needs to backtrack to the correct location. With the navigation steps on the internal route $\boldsymbol{\tau}'$ generated through the graph, the agent keeps updating the hidden states $\boldsymbol{h}$ based on the observation $\boldsymbol{o}_t$ from the environments.

**Supervising imitation learning** The proper training of the imitation learning agent has been a challenging problem [44, 41, 43]. Student forcing with new computed route as supervision is a widely used solution in navigation [14, 19, 2]. However, the mismatch between the new route and language instructions could potentially lead to noisy and incorrect signals for learning language groundings and navigation. We provide a new graph-augmented solution for computing the supervision for each student sampled trajectory. Assume a metric $\mathcal{M}(\cdot, \cdot)$ between two navigation trajectories (e.g. [46]). With a graph memorizing the entire possible action space, the subset of nodes in ground

truth route $\boldsymbol{\tau}^*$ is guaranteed to exist in $\mathcal{G}_t$. We choose the node $\boldsymbol{v}$ on the $\boldsymbol{\tau}^*$ that maximizes the metric $\mathcal{M}(\boldsymbol{\tau}_t \cup (\boldsymbol{v}), \boldsymbol{\tau}^*)$ as the ground truth action for step $t$. This provides a "correction" signal that indicates the best action to take for correcting the mistake made by the agent. Since this supervision comes from ground truth trajectories, it is free of noisy signals that mismatches the instruction and does not require additional environment information to re-compute the new shortest-paths.

## 3.2 Evolving Graphical Planner: Planning Core

With the graphical representation $\mathcal{G}_t$, a straightforward method is to directly perform planning[1] on the full graph using the embeddings of nodes and edges. However, a progressively growing graph can lead to high costs and limit the scalability of the model. Pre-exploration based methods often tackle this issue through performing offline sparsification with multiple rounds of cleanups on the pre-collected graphs [13] containing full knowledge over the map, which are unsuitable under the online navigation settings. In this section, we show that, interestingly, the effective planning can be achieved not through the full graph, and present the second component of our model, where it performs a goal-driven information extraction dynamically on the entire graph $\mathcal{G}_t$ to build a condensed proxy representation $\bar{\mathcal{G}}_t$ used for planning. Our model utilizes Graph Neural Networks (GNN) as a basic operator, which we explain at the end of the section.

**Proxy graph** Denote the proxy graph as $\bar{\mathcal{G}}_t = (\bar{\mathcal{V}}_t, \bar{\mathcal{E}}_t, \bar{\boldsymbol{M}}_t)$, where $\bar{\mathcal{V}}_t = \{\bar{\boldsymbol{v}}^1, \bar{\boldsymbol{v}}^2, ..., \bar{\boldsymbol{v}}^{\bar{N}}\}, \bar{\boldsymbol{v}}^i \in \mathbb{R}^d$, $\bar{\mathcal{E}}_t = \{\bar{e}^{ij}\}$ and $\bar{\boldsymbol{M}}_t$ are the pooled node embedding set, edge embedding set and connectivity matrix with function types respectively. $\bar{N}$ is the size of proxy graphs and remains fixed for all steps. The proxy graph $\bar{\mathcal{G}}_t$ contains a fixed number of nodes invariant to the growing graph size in $\mathcal{G}_t$. We hypothesize that given the instruction information and the current states of the agent, there are only a subset of nodes providing useful information for planning. To construct the proxy representation, the model uses a normalized pooling similar to [47]. Differently, our graphical representation consists of a rich set of information including edge states, function types in connectivity matrices besides the node states. We describe the process of generating the proxy representation and corresponding planning as follows.

Given the entire graphical representation $\mathcal{G}_t = (\mathcal{V}_t, \mathcal{E}_t, \boldsymbol{E}_t)$, the planner contains two functions, $\text{GNN}_L$ and $\text{GNN}_P$. The function $\text{GNN}_L$ takes in the agent state information and constructs the proxy representation through a lightweight neural network. Assume the pooling function uses $d_L$ as the graph dimension in the propagation model, and performs $K_L$ step message passing. We derive the pooling matrix as follows, using a small $d_L$:

$$\boldsymbol{A}_t \quad = \quad \text{softmax}\Big(\text{GNN}_L\big(\mathcal{G}_t, [\boldsymbol{h}_t; \boldsymbol{c}]; K_L, d_L\big)\Big) \in \mathbb{R}^{|\mathcal{V}_t| \times |\mathcal{V}'_t|}$$

The normalized pooling matrix is the attention weights on the entire graph and extracts relevant information conditioned on the agent states $\boldsymbol{h}_t$ and instruction embeddings $\boldsymbol{c}$ for further planning. To help the description, we denote the concatenated matrix of all node vectors as $\boldsymbol{V}_t \in \mathbb{R}^{|\mathcal{V}_t| \times d}$, the concatenation of all edge vectors as tensor $\boldsymbol{E}_t \in \mathcal{R}^{|\mathcal{V}_t| \times |\mathcal{V}_t| \times d}$. The concatenation order is aligned with the connectivity function matrix. The following operations are used for deriving the proxy representation.

$$\bar{\boldsymbol{V}}_t \quad = \quad \boldsymbol{A}_t^T \boldsymbol{V}_t \in \mathbb{R}^{|\mathcal{V}'_t| \times d}$$
$$\bar{\boldsymbol{M}}_{t,i} \quad = \quad \boldsymbol{A}_t^T \boldsymbol{M}_{t,i} \boldsymbol{A}_t \in \mathbb{R}^{|\mathcal{V}'_t| \times |\mathcal{V}'_t|}, i \in \{1, 2, ..., |\mathcal{F}|\}$$
$$\bar{\boldsymbol{E}}_{t,i} \quad = \quad \boldsymbol{A}_t^T \boldsymbol{E}_{t,i} \boldsymbol{A}_t \in \mathbb{R}^{|\mathcal{V}'_t| \times |\mathcal{V}'_t|}, i \in \{1, 2, ..., d\}$$

where $\bar{\boldsymbol{M}}_{t,i}$ is a non-negative matrix indicating the weights of connectivity among nodes for function type $i$, $\bar{\boldsymbol{E}}_{t,i}$ is the matrix at $i_{th}$ dimension for the edge state tensor. The pooled tensors $\bar{\boldsymbol{V}}_t$ and $\bar{\boldsymbol{E}}_t$ are corresponded to the node set $\bar{\mathcal{V}}_t$ and edge set $\bar{\mathcal{E}}_t$ of the proxy graph $\bar{\mathcal{G}}_t$

**Planning** The planning of the navigation agent is achieved through propagating information among nodes in the proxy graph, conditioned on the agent state information $\boldsymbol{h}_t$ and instruction encodings $\boldsymbol{c}$. Denote the graph dimension used in the propagation model as $d_P$, the number of steps for

message passing operations as $K_P$, the refined node embedding of the proxy graph is derived through $\text{GNN}_P, (\mathcal{V}_t, [\boldsymbol{h}_t; \boldsymbol{c}]; K_P, d_P)$, with $d_P$ controlling the capacity of the function. The refined node embedding contains the information involving the neighboring nodes (visited locations and unexplored actions), the state of agent, the instruction, and the connectivity types between nodes. With the globally refinement step, node embedding vector is unpooled back to the original graph through $\boldsymbol{V}_t = \bar{\boldsymbol{V}}_t \boldsymbol{A}_t^T$. With the update node representation containing both the current state of the agent and the full action-observation space in the history, the distribution over actions is generated based on the node vectors:

$$\hat{a}_t^i = f_{\text{out}}(\boldsymbol{h}_t, \boldsymbol{v}_t^i); p(a_t^i | \boldsymbol{h}_t, \boldsymbol{V}_t) = \exp(\hat{a}_t^i) / \sum_j \exp(\hat{a}_t^j) \tag{1}$$

where the $f_{\text{out}}$ function is a dot-product with linear mapping $(W_{hv}h_t)^T \boldsymbol{v}_t^i$ parameterized by $W_{hv}$.

**Multi-channel planning** The model is further strengthened with the ability to perform multi-channel planning over the graph $\mathcal{G}_t$. Instead of only using one proxy graph representation $\bar{\mathcal{G}}_t$ to extract information and perform propagation, we find it useful to learn a set of proxy graphs $\{\bar{\mathcal{G}}_t\}$ and perform planning independently on each of them. The final information are aggregated through summation over the embedding across the channels. The final policy over actions is generated through the same operation described in Eqn. 1

**Training objective** We train our full agent through the standard maximum likelihood objective using cross-entropy loss. Given the demonstration $\mathcal{D}$, the loss function to optimize is formulated as:

$$\mathcal{L} = \mathbb{E}_{(\boldsymbol{\tau}^*, \boldsymbol{x}, \text{ENV}) \sim \mathcal{D}} \Big[ \sum_t -y_t^i \log(p(a_t^i | \boldsymbol{h}_t, \boldsymbol{c} = \text{LSTM}(\boldsymbol{x}))) \Big] \tag{2}$$

where $\boldsymbol{\tau}^*$ is the ground truth trajectory, $y_t^i$ is the ground truth label on action $i$ at step $t$, generated through the graph-augmented supervision using the information of $\boldsymbol{\tau}^*$, as described in Sec. 3.1.

### 3.2.1 Message Passing Operations

In this subsection we explain the Graph Neural Network (GNN) operator used in the EGP. As the operator is used in both pooling and planning, we describe it as a general function which takes in a graph $\mathcal{G}$ and a general context information $\boldsymbol{r}$ (e.g., the agent hidden state and language encodings), with hyper-parameters $K$ and $d_{\mathcal{G}}$. Formally, given the input graph $\mathcal{G} = (\mathcal{V}_0, \mathcal{E}_0, \boldsymbol{M})$ and the context vector $\boldsymbol{r}$, the function $\text{GNN}(\mathcal{G}, \boldsymbol{r}; K, d_{\mathcal{G}})$ generates the refined node vectors $\mathcal{V}_{K+1}$ after $K$ steps of message passings, where $d_{\mathcal{G}}$ is the vector dimension in the propagation model. The subscription on nodes and edges denotes the index of message passing iterations, with a slight abuse of notation. The $\text{GNN}(\cdot, \cdot; K, d_{\mathcal{G}})$ function contains two components: an input network and a propagation network.

**Input network** Along with the initial vectors for nodes and edges, the input network considers the context vector $\boldsymbol{r}$ as a shared additional information across nodes and edges. The input model maps the context vector with node and edge vectors respectively into two fixed-size embedding as follows:

$$\boldsymbol{v}_1^i = f_{\text{in}}(\boldsymbol{v}_0^i, \boldsymbol{r}); \boldsymbol{e}_1^{ij} = f_{\text{in}}(\boldsymbol{e}_0^{ij}, \boldsymbol{r}); \boldsymbol{v}_1^i, \boldsymbol{e}_1^{ij} \in \mathbb{R}^{d_{\mathcal{G}}} \tag{3}$$

where $f_{\text{in}}$ is a neural network, and the generated embedding vectors are used for message communications among graph nodes in the propagation model.

**Propagation network** The propagation model, taking in the mapped embedding vectors $\{\boldsymbol{v}_1^i\}$, consecutively generates a sequence of node embedding vectors $\boldsymbol{v}_k^i, k \in \{1, ..., K+1\}$ for each node. At step $k$, the propagation operation updates every node through computing and aggregating information from the neighborhood nodes. The process is executed in the order:

$$\boldsymbol{e}_k^{ij} = \sum_n \boldsymbol{M}_{i,j,n} \cdot f_n\Big(\boldsymbol{v}_k^i, \boldsymbol{v}_k^j, \boldsymbol{e}_1^{ij}\Big) \in \mathbb{R}^{d_{\mathcal{G}}}; \tag{4}$$

$$\boldsymbol{v}_{k+1}^i = g\Big(\sum_{j \in \mathcal{N}_i} \boldsymbol{e}_k^{ij}, \boldsymbol{v}_k^i\Big) \in \mathbb{R}^{d_{\mathcal{G}}} \tag{5}$$

where $f_n(\cdot, \cdot, \cdot) : \mathbb{R}^{3d} \to \mathbb{R}^d$ is the message function. Function $g(\cdot) : \mathbb{R}^{2d} \to \mathbb{R}^d$ is the aggregator function that collects messages back to node vectors. $\mathcal{N}_i$ represents the neighbours of node $i$ in the graph. The refined node vector set $\mathcal{V}_{K+1} = \{\boldsymbol{v}_{K+1}^i\}$, containing global information from the whole graph is mapped through a matrix $W_{\text{out}}$ to recover the input node dimension $d_{\text{in}}$ and is used as the output of the $\text{GNN}(\cdot, \cdot)$ function for either pooling or planning component, as described in Sec. 3.2.

# 4 Experiments

## 4.1 Experimental setup

**Datasets** We evaluate our method on the standard benchmark datasets for Vision-and-Language Navigation (VLN). The VLN task is built upon photo-realistic simulated environments [48] with human-generated instructions describing the landmarks and directions for navigation routes. Starting at a random sampled location in the environment, the agent needs to

| Dataset | Train | Val:seen | Val:unseen | Test |
|---------|-------|----------|------------|------|
| R2R | 14,039 | 1,021 | 2,349 | 4,173 |
| R4R | 233,532 | 1,035 | 45,234 | - |

Table 1: Dataset statistics.

follow the instruction to navigate through the environment. There are two datasets commonly used for VLN: (1) Room-to-Room (R2R) benchmark [1] with 7,189 paths, each associated with 3 sentences, resulting in 21,567 total human instructions. The paths are produced through computing shortest paths from start to end points; (2) Room-for-Room (R4R) [18], which extends the R2R dataset by re-emphasizing on the necessity of following instructions compared to the goal-driven definition in R2R. The R4R dataset contains 278,766 instructions associated with twisted routes connecting two shortest-path trajectories in R2R. The dataset details are summarized in table 1.

**Implementation details** We follow [14] and adopt the co-grounding agent (w/o auxiliary loss) as our base agent. As the standard protocol for R2R, visual features for each location are pre-computed ResNet features from the panoramic images. In the Evolving Graphical Planner, we use 256 dimensions as the graph embedding size for both the full graph and the proxy graph. The propagation model uses three iterations of message passing operations. For every expansion step, the default setting adds all the possible navigable locations into the graph (top-K is set to 16, the maximum number of navigable location in both datasets). For student-forced training, we use graph-augmented ground truth supervision throughout the experiments except for the ablation study on supervision methods. The model is trained jointly, using Adam [49] with 1e-4 as the default learning rate.

## 4.2 Room-to-Room benchmark

**Evaluation metrics** We follow prior works on the R2R dataset and report: navigation error (NE) in meters, lower is better; Success Rate (SR), i.e., the percentage of navigation end-locations which are within 3m of the true global location; Success Rate divided by path Length in meters (SPL); and Oracle Success Rate (OSR), i.e., whether the path at any point passes within 3m of the goal state.

### 4.2.1 Comparison with prior art

**Architectures for comparison** We compare our model with the following state-of-the-art navigation *architectures*: (1) the Seq2Seq agent [1] that translates instructions to actions; (2) Speaker-Follower (SF) [2] agent that augments the dataset with a speaker model; (3) the Reinforced Cross-Modal (RCM) agent [50] using modal-alignment score as intrinsic reward for reinforcement learning; (4) the Self-Monitoring (Monitor) agent [14] that uses a co-grounding module and a progress estimation component to increase the progress alignment between texts and trajectories; (5) the Regretful agent [19] that uses the Regretful module and Progress Marker to perform one-step rollback; (6) the Ghost [27] with Bayesian filters.

**Results** We report results in table 2. We train our models both by using only the standard demonstration and by augmenting the dataset with the synthetic data containing 178,330 instruction-route pairs generated by the Speaker model [2]. On the Val Unseen split, we observe that just through using the EGP module (without synthetic data augmentation), the performance of agent can be increased over the baseline agent by $0.86$ meters on NE (from $6.20$ to $5.34$), by $9\%$ on SR (from $43\%$ to $52\%$), by $5\%$ on SPL (from $36\%$ to $41\%$), and by $13\%$ on OSR (from $52\%$ to $65\%$). Our path length remains short, at 13.7 meters compared to 12.8m for baseline, 14.8m for RCM [50] and 15.2m SF [2] (not shown in the table). Most notably, our EGP agent with synthetic data augmentation outperforms prior art on all metrics, across both the validation-unseen and the test set. Concretely, on the test set we achieve a $0.35$ meters reduction in NE (from $5.69$ to $5.34$), a $5\%$ improvement on SR (from $48\%$ to $53\%$), a $2\%$ improvement on SPL (from $40\%$ to $42\%$), and a $2\%$ improvement on OSR (from $59\%$ to $61\%$) over the best performing prior model on each metric respectively.

| Models | Type | Val Unseen | | | | Test | | | |
|---|---|---|---|---|---|---|---|---|---|
| | | NE ↓ | SR$^\%$ ↑ | SPL$^\%$ ↑ | OSR$^\%$ ↑ | NE ↓ | SR$^\%$ ↑ | SPL$^\%$ ↑ | OSR$^\%$ ↑ |
| Seq2Seq [1] | IL | 6.01 | 39 | - | 53 | 7.81 | 22 | - | 28 |
| Ghost [27] | IL | 7.20 | 35 | 31 | 44 | 7.83 | 33 | 30 | 42 |
| SF* [2] | IL | 6.62 | 36 | - | 45 | 6.62 | 35 | 28 | 44 |
| RCM* [50] | IL+RL | 5.88 | 43 | - | 52 | 6.12 | 43 | 38 | 50 |
| Monitor [14] | IL | 5.98 | 44 | 30 | 58 | - | - | - | - |
| Monitor* [14] | IL | 5.52 | 45 | 32 | 56 | 5.67 | 48 | 35 | 59 |
| Regretful [19] | IL | 5.36 | 48 | 37 | 61 | - | - | - | - |
| Regretful* [19] | IL | 5.32 | 50 | 41 | 59 | 5.69 | 48 | 40 | 56 |
| Fast* [15] | IL | 4.97 | 56 | 43 | - | 5.14 | 54 | 41 | - |
| Baseline agent | IL | 6.20 | 43 | 36 | 52 | - | - | - | - |
| **EGP (ours)** | IL | 5.34 | 52 | 41 | **65** | - | - | - | - |
| **EGP* (ours)** | IL | **4.83** | **56** | **44** | 64 | **5.34** | **53** | **42** | **61** |

Table 2: We compare our architecture with previous state-of-the-art architectures on the *Val Unseen* and *Test* splits of R2R [1]. (∗: models using additional synthetic data. %: numbers in percentage). We also compare our results with a graph-search based post-processing method [15] which uses longer path-length, hand-crafted rules and other information such as progress monitor and speaker scores.

**Discussion of other works** Note that there are other works contributing to this benchmark through *non-architecture* approaches: using more data (6,582K) for BERT-type pre-training [4]; exploiting web data [51]; adding extra tasks [52]; adding dropout regularization [53]; different settings of evaluation (fusing information from three instructions) [54, 55]; post-processing decoding method [15]. We contribute a clean and differentiable backbone navigation architecture and these works can potentially be useful as complementary approaches.

### 4.2.2 Ablation studies

We now justify the design choices of our model by analyzing the individual components. In addition to the metrics above we also include Path Length (PL) for completeness. Results are summarized in table 3, with the last row depicting our model with the default settings.

**Global planning** We verify the importance of global planning through controlling the top-K expansion rate for the graphical representation. In R2R dataset, there are maximally 16 navigable locations for each state. With a smaller expansion rate, the EGP planner has less expressive power on exploiting global information from the environments. As seen in the top group of rows of table 3, with smaller top-K, the path becomes shorter (fewer options to explore) and the accuracy of the model consistently drops, indicating the importance of global planning.

**Planner implementation** Next we analyze the effects of message passing (mp) steps and the number of channels used on our planner module. The results are summarized in the second group of rows in table 3. Through the information propagation operations, our model achieves a 8% increase on SR (from 42% with mp=0,channel=1 to 50% with mp=3,channel=1).With more independent planning channels, we can obtain a further 2% improvement on SR (from 50% with mp=3,channel=1 to 52% for our default setting with mp=3,channel=3, last row). To verify whether the increase is due to more parameters in the model, we also add a comparison through using a single channel planner with three times larger graph dimensions (768), which shows no similar effect to the multi-channel models.

**Supervision methods** Finally, we compare our method a navigation agent trained by the standard student forcing. The standard supervision used for student forcing requires recomputing the new shortest-path route to the goal from each location, leading to potential noisy data and larger generalization error, shown in the second-from-last row of table 3.

### 4.3 Room-for-Room benchmark

**Evaluation metrics** The Room-for-Room dataset emphasizes on the ability of correctly following instructions instead of solely on reaching the goal locations. We follow the metrics in [18, 46] and

| Ablation type | Model | PL$\downarrow$ | NE$\downarrow$ | SR$^\%\uparrow$ | SPL$^\%\uparrow$ | OSR$^\%\uparrow$ |
|---|---|---|---|---|---|---|
| Global vs local planning | EGP - topK = 3 | **13.07** | 5.95 | 47 | 38 | 56 |
| | EGP - topK = 5 | 13.17 | 5.75 | 49 | 40 | 59 |
| | EGP - topK = 10 | 13.50 | 5.71 | 50 | 40 | 61 |
| Planner implementation | EGP - mp=0,channel=1 | 18.83 | 6.06 | 42 | 32 | 62 |
| | EGP - mp=3,channel=1 | 14.65 | 5.73 | 50 | 40 | 62 |
| | EGP - graph dim $\times$ 3 | 14.16 | 5.68 | 49 | 38 | 60 |
| Supervision | EGP - with shortest path | 14.68 | 5.65 | 46 | 36 | 57 |
| – | EGP | 13.68 | **5.34** | **52** | **41** | **65** |

Table 3: We show ablation studies of our method on the val-unseen set of Room-to-Room. The default setting for our model (bottom row) is top-K= 16, mp=3, channel=3, graph dimension=256; we use graph-augmented supervision rather than shortest path for training.

mainly compare our results on Coverage weighted by Length Score (CLS) that measures the fidelity of the agent's path to the reference, weighted by the length score, and the Success rate weighted normalized Dynamic Time Warping (SDTW) and normalized Dynamic Time Warping (nDTW) that measure the spatio-temporal similarity of the paths by the agent and the expert reference.

**Architectures for comparison** We compare our model with the Speaker-Follower model [2], Reinforced Cross-Modal agent [50] trained under goal-directed and fidelity-oriended rewards, reported in [18], and the Perceive Transform Act (PTA) agent [56] using more complex multi-modal attentions.

**Results analysis** We summarize the results in table 4. Note that all previous state-of-the-art methods require the mixed training between imitation learning and reinforcement learning objectives. The student forcing method leads to goal-oriented supervisions and harms the ability of following instructions for agents [18]. Our model is the first that successfully train the navigation agent through pure imitation learning on the R4R benchmark, due to the benefit of the graphical representation, the powerful planning module and the new supervision method. We obtain a consistent margin across all metrics. Specifically, our model outperforms other architectures by 7.0, 5.0 and 4.9 on the fidelity-oriented measurements CLS, nDTW and SDTW respectively. Also, although using a global search mechanism, our model maintains a relatively short path length, which is difficult in other rule-based global search algorithms [2, 15].

| Models | Type | PL | NE$\downarrow$ | SR$^\%\uparrow$ | CLS$\uparrow$ | nDTW$\uparrow$ | SDTW$\uparrow$ |
|---|---|---|---|---|---|---|---|
| Random | - | 23.6 | 10.4 | 13.8 | 22.3 | 18.5 | 4.1 |
| Speaker-Follower[18] | IL+RL | 19.9 | 8.47 | 23.8 | 29.6 | - | - |
| RCM + goal-oriented[18] | IL+RL | 32.5 | 8.45 | 28.6 | 20.4 | 26.9* | 11.4* |
| RCM + fidelity-oriented[18] | IL+RL | 28.5 | 8.08 | 26.1 | 34.6 | 30.4* | 12.6* |
| PTA low-level[56] | IL+RL | 10.2 | 8.19 | 27.0 | 35.0 | 20.0 | 8.0 |
| PTA high-level[56] | IL+RL | 17.7 | 8.25 | 24.0 | 37.0 | 32.0 | 10.0 |
| **EGP (ours)** | IL | 18.3 | **8.0** | **30.2** | **44.4** | **37.4** | **17.5** |

Table 4: Comparison across methods on R4R Val Unseen split. Path Length (PL) is reported as a reference. (*Note that we refer to the numbers from [46] as [18] did not report DTW-based results)

## 5  Conclusion

In this work, we proposed a solution to the long-standing problem of contextual global planning for vision-and-language navigation. Our system based on the new Evolving Graphical Planner (EGP) module outperforms prior backbone navigation architectures on multiple metrics across two benchmarks. Specifically, we show that building a policy over the global action space is critical to decision making, the graphical representation can further elicit a new supervising strategy for student forcing in navigation, and, interestingly, the actual planning can be achieved through a proxy graph rather than the actual topological representation which leads to high costs in both time and memory.

## Broader Impact

This work has several downstream applications in areas like autonomous navigation and robotic control, especially through the use of natural language instruction. Potential downstream uses of such agents range from healthcare delivery to elderly home assistance to disaster relief efforts. We believe that imbuing these agents with a global awareness of the environment and long-term planning will enable them to handle more challenging tasks and recover gracefully from mistakes. While the graph-based approach we propose is scalable and easy to manipulate in real time, future research can address computation challenges and increase the planning time-scale to enable better decision making.

## Acknowledgments and Disclosure of Funding

This work is partially supported by King Abdullah University of Science and Technology (KAUST) Office of Sponsored Research (OSR) under Award No. OSRCRG2017-3405 and by Princeton University's Center for Statistics and Machine Learning (CSML) DataX fund. We would also like to thank Felix Yu, Angelina Wang and Zeyu Wang for offering insightful discussions and comments on the paper.

## Footnotes

[1]The planning term here refers to an implicit process using message passings to maximize a fitness metric (such as nDTW [46] imposed through $\mathcal{M}(\cdot, \cdot)$), compared to other methods that explicitly maximize a fitness score to perform planning.

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
