[Supplementary Material]

# Supplementary material

**Zhiwei Deng**    **Karthik Narasimhan**    **Olga Russakovsky**
Department of Computer Science
Princeton University
{zhiweid, karthikn, olgarus}@cs.princeton.edu

## 1  Navigation model details

We here provide more implementation details for our models, including the base navigation agent and the Evolving Graphical Planner module.

### 1.1  Base agent details

We follow the details in [1] and adopts the co-grounding module for matching visual observations and textual instructions. For every step, the agent receives the observation in the form of a panoramic image, following [2]. Each action is a navigable location to teleport to, and is described through the pre-computed ResNet-152 features and corresponding angle information [2]. The agent keeps the hidden states $h_t$ through an LSTM model. For both the language encoding LSTM and the agent LSTM, the hidden dimension is set to 1024. Please refer to https://github.com/chihyaoma/regretful-agent for more details.

### 1.2  Evolving Graphical Planner details

Our Evolving Graphical Planner serves as an additional module to perform contextual planning across actions. The planner initializes the graph node features through the pre-computed features. The visited node contains a panoramic image and the corresponding feature is derived through a mean pooling operation. The action node contains the raw feature from the corresponding image and the angle information. For both action nodes and visited nodes, the feature is further processed through an MLP with BarchNorm operations. The planner concatenates the language encoding $c$ and agent states $h_t$ with the node features to acquire the contextual information. The final node information $\{v_t\}$ is planned through several steps of message passing operations. To obtain the policy over navigable actions, we follow [2] and use soft-attention to compute similarities between the agent's information $\hat{h}_t = W_{h\hat{h}} h_t$ and the action nodes in the graph.

### 1.3  The information used in Room-to-room

There has been works indicating that a navigation agent trained in the Room-to-room benchmark can simply rely on angle and instruction information to perform navigation [3]. This is potentially due to the sparse topological connections used for defining the room structures. We further investigated the information sources that are used in our navigation agent. First, we remove the language information (instructions) in EGP's input. Compared to the full model SR (52%), without language, EGP has 28% SR, showing that EGP is best used with language and is indeed utilizing the language information. Second, we remove the vision feature used in EGP and obtained 47% SR, indicating EGP inherits the angle domination phenomenon in this benchmark. Note that EGP is a general navigation module designed for the purpose of expanding the decision space and making global differentiable planning. The modality issue can be potentially alleviated through the ensemble technique proposed in [3].