[Reviews · NeurIPS 2020]

Review 1

Summary and Contributions: This paper presents a graph-based navigation approach that can be applied to vision-and-language navigation (VLN) tasks which are situated in a discrete navigation graph, e.g. the benchmark Room-to-Room dataset. Most past work on VLN makes local decisions at each timestep, choosing which of the nodes directly connected to the agent's current node in the navigation graph to visit next. In contrast, this approach chooses the node to visit next from a global set of fringe nodes which are connected to any previously visited node. To select which node to visit, the approach uses two layers of graph neural networks: a base graph network, which summarizes observation information over all the navigation graph nodes that have been visited so far, and a proxy graph, which is designed to be sparser, operating on a fixed number of (I believe) abstract vertices, with vertex representations pooled from the underlying base graph network. These abstract vertex representations are projected back to the navigation graph structure to select a next node to move to.

Strengths: The model obtains strong results on two different benchmark datasets, R2R and R4R, and carries out a series of ablations showing the effects of most components introduced in the model. The model architecture is novel for the VLN task, and the idea of scoring nodes in the frontier of visitable nodes was well-motivated as a way to do goal-driven planning (at least under a task framing that allows using the navigation graph structure in this way).

Weaknesses: While the paper uses a form of graph search (maintaining a set of scored possible nodes that can be traversed to, and backtracking through the graph to reach them), if I understood correctly it doesn't compare to the graph search versions of other models from past work. Please see relation to prior work. As currently written, enough details were unclear about the model (in particular the proxy graph) and training, and the evaluation that I suspect that even researchers working on language grounding tasks that would be amenable to this type of graph search would be limited in their ability to apply and build on the model in the paper.

Correctness: I was unclear on whether the evaluation metrics were computed taking path length into account, which is important for comparison to past work. Do the path lengths used to calculate the evaluation metrics (SPL and OSR both depend on path lengths) incorporate the steps taken on the internal route generated by backtracking through the graph from the agent's current location to reach a node in the "potential action nodes"? Line 155, which says that the hidden state is updated, suggests this; however the supplementary ("a navigable location to teleport to") and the very high SPL scores suggest that it might not be incorporated. Path length has a large effect on SPL scores, and the SPL scores here are much higher than (to my knowledge) any other VLN method that requires backtracking through visited nodes (the graph search methods in Fried et al., Ma et al., and Ke et al.).

Clarity: The "supervising imitation learning" section of 4.1 was unclear to me. Since the graph expands dynamically from an initially empty graph, how is it guaranteed that the "subset of nodes in ground truth route \tau_t^* is guaranteed to exist in G_t"? How is a connected route created from the set of vertices \tau_t \cup (v)? How does this fix the mismatch between new route and language instructions mentioned in lines 160-161, as it seems that mistakes can still result in the agent exploring nodes that don't match the language instructions? I was also unclear on the "proxy graph" section of 4.2. This section describes a fixed number of nodes which are invariant to the growing graph size. Since they are fixed while the underlying graph is expanding dynamically, it would be helpful to give some intuition or analysis of what roles these nodes play, e.g. how many nodes are there, do they have a spatial interpretation, and does the connectivity given by the attention pooling matrix A_t change substantially over time. How is the "EGP - with shortest path" model in Table 3 trained? I found the claim "The student forcing method leads to goal-oriented supervisions and harms the ability of following instructions for agents" to be unclear. How does this approach, which also uses student forcing, fix this issue?

Relation to Prior Work: It seems that all the results in Table 2 for prior work are with models that perform greedy inference (i.e. no graph search). Since this paper uses graph search, it would be more fair to compare to the graph search variants of past work. When using search, the SF and Monitor models obtain SRs of 53% and 61% respectively -- although very low SPLs of 1% and 2% respectively -- and the FAST method of Ke et al. (which doesn't seem to be compared to in the table) obtains SR/SPL scores of 54% and 41% without search, or 61% and 3% with search. All these SR scores are at least as good as the 53% reported here; and while this paper has a much higher SPL I have questions about whether this was correctly computed (see correctness). The paper should to be clearer about the relationship and comparison to other search-based methods which also uses a frontier-based method of selecting the next node to visit in the graph. This paper characterizes those methods as "rule-based" or "post-processing" but it's unclear to me what this means since those (like this paper) use a neural parameterization of a scoring function and then run a graph search algorithm on top of these scores. The major difference seems, to me, to be in the architecture of the neural network used to parameterize the scoring function (this work uses graph neural networks, other work uses seq2seq models) and the sophisticated form of student forcing used in this paper to train the node selection method that matches the search procedure used (other work uses only models trained without search to rescore, or in the case of Ke et al. use a reranker trained on candidates output by search). I think these are all valid and substantial contributions for this paper to make, but they should be more clearly compared. It would also be helpful to give more explanation of how this approach compares to the end-to-end differentiable belief-state approach of Anderson et al.

Reproducibility: No

Additional Feedback: 148: "ranked by confidence scores in policy distribution": are these nodes with the highest probability under the current policy \pi? 163: I was unclear on what metric M is used. I think it would be helpful to give some more motivation for the proxy graph. Is the proxy graph necessary because the underlying navigation graphs are too large or too densely connected to run a GNN on? Or does the proxy graph improve performance on the metrics? Can "mp" in Table 3 be defined in terms of the K_p or K_L defined in section 4.2? --- update after author response --- Thank you for all your helpful clarifications! They largely address my concerns, and I've updated my score to a 6. Supervising strategy: Thanks, this was very helpful. My confusion here was that since the agent can explore, entire routes that it follows may not correspond to the language instructions (a property of any student forcing method). But as you point out -- and I overlooked -- the method does clearly have the advantage that following the oracle from any given location the agent has reached will only require it to go through nodes in the GT (a property previous student forcing setups lack). Adding some of the "mismatch" text from the response to the paper might help clarify this to others who were confused as I was. Changing "GT action" -> "GT node" in places may also help reinforce this, as GT action may convey (to people more familiar with previous VLN setups) an edge in the navigation graph (i.e. require starting in a GT node) -- although this should be possible to figure out from the paper's method. Thanks for the assurances about the computation of the path length in backtracking, and the explanation of the shorter path lengths due to the novel supervision method here. It would help to emphasize these more in the paper. Proxy graph: Thanks for these clarifications! I think it would strengthen the motivation to mention the R4R efficiency (although the ablation on R2R is also helpful to know): the same efficiency issues will also likely be present in other big navigation graphs like Touchdown, and the proxy graph could be a promising method to scale this EGP approach to them. Re: greedy inference, I do appreciate that the method is only maintaining one complete path of past actions. But since it chooses an extension node from a set of nodes (as beam search / graph search methods do) which aren't locally-connected to the agent's current node, I still feel that the paper should include a comparison to graph search methods. It's a great strength of this paper that it uses an elegant approach (the training procedure matches the inference procedure) and achieves higher SPL than that past work while not using many of the components (as the paper and rebuttal point out).


Review 2

Summary and Contributions: In this paper, the authors introduce the Evolving Graphical Planner (EGP), which is a model performing global planning for navigation based on raw sensory input. The model can dynamically construct a graphical representation and generalize the action space to allow for more flexible decision making and perform efficient planning on a proxy graph representation. The proposed approach is tested on two benchmark datasets, which shows state-of-the-art performance.

Strengths: 1. The paper is well organized and easy to follow. The structure is clear and language is good. 2. The vision-and-language navigation topic is interesting and related to the NeurIPS community. 3. The authors introduced a novel method for training the planning modules with imitation learning, which allows the agent to efficiently learn how to select global actions and backtrack when necessary. They also introduce proxy graphs, which allows for more scalable planning. The novelty is significant. 4. The proposed method is validated on two benchmark datasets and comprehensive quantitative and ablative studies are illustrated to support the claims.

Weaknesses: 1. The authors provided comprehensive quantitative and ablative results and analysis. However, it would be better to also present qualitative results such as videos to demonstrate the advantages of the proposed method.

Correctness: No major mistake was found. The claims can be validated by the empirical results in the paper.

Clarity: The paper is well organized and easy to read. The structure is clear and language is good.

Relation to Prior Work: The authors list some challenges of the vision-and-language navigation task in the introduction section. They also pointed out that most existing works just deal with a subset of the challenges or requires additional pre-processing steps, while the proposed approach is more flexible. In the related work section, the authors listed a few related research and pointed out their weakness.

Reproducibility: Yes

Additional Feedback: Thanks for the authors' response. After reading the rebuttal, I would like to keep my original score.


Review 3

Summary and Contributions: The paper introduces a new method for vision and language navigation in unseen environments. At the core of the approach is a connectivity-graph that is expanded as the agent navigates the world and encounters new navigable nodes. From this graph, EGP builds multiple proxy-graphs by computing an attention matrix over the graph nodes, and projecting the graph according to this matrix. Information is propagated within the proxy graphs using a graph neural network, before reconstructing a graph that aligns with the original spatial graph. Finally, the node vectors are used to predict probabilities that the node should be navigated (action probabilities). The core contributions are: 1. A graphical planning algorithm for the VLN task that relies on graph neural networks. 2. A method for pooling information from a neural graph representation to obtain a sparse graph for more efficient planning.

Strengths: * The approach is very technically interesting and generally applicable to navigation domains that have an underlying graphical connectivity structure, which have seen great interest in the community recently. * The empirical results are useful to the VLN community, in that they inform them of the performance of a GNN-based approach that is appropriate for the task. * Useful ablations help the reader understand the source of the method's performance. * The approach has the potential to replace explicitly hand-engineered decoding schemes, such as beam search in VLN.

Weaknesses: * Two important ablations are missing (discussed below). Without these, it is not clear that the approach is actually grounding language in visual observations. * The approach assumes an underlying graphical structure, and so would not be directly applicable to continuous natural language navigation domains that more closely reflect real-world robot applications (e.g. Habitat [Savva et. al., 2019], Lani [Misra et. al., 2018; Blukis et. al. 2019]). Deployment in such environments would require additional connectivity prediction.

Correctness: * The positioning of the work as a planner is confusing, and casts doubt on the claims that it outperforms state-of-the-art approaches. For example, the Tactical Rewind approach that presents a decoding scheme that relates to planning is excluded from empirical comparison due to being "orthogonal", but outperforms the presented approach. * The no-language and no-vision ablations should be added. It has been shown in the past (see https://arxiv.org/pdf/1906.00347.pdf) that the graph structure communicates a lot of information and allows achieving good performance by ignoring vision. This is of particular concern with an approach that encodes the graph using a GNN.

Clarity: The paper is well written, and the approach is overall well presented. It includes a comprehensive discussion of related work, and introduces preliminaries in a way that allows easily understanding the technical approach. For example, I had only a very superficial awareness of graph neural networks, but was brought up to speed well enough to understand the technical material in the paper. However, the positioning of the approach as a planner was not clear. In sections 1-4, the approach is presented as a planner, whereas in the results section, work that could be interpreted as planning (Tactical Rewind) is omitted as being orthogonal.

Relation to Prior Work: Overall, yes. The authors have discussed relationship to most relevant work in vision-and-language navigation. However, there is other relevant work on navigation instruction following in environments that are not graph-based that the authors did not address (e.g. Misra et. al., 2018).

Reproducibility: Yes

Additional Feedback: * This is by no means a critique of the approach, but It wasn’t clear to me what is the connection between the proposed method and planning. Planning (in AI) typically refers to the process of finding a sequence of actions that result in behavior that achieves a goal, typically as measured by a fitness metric. In this case, the model is trained directly with action supervision via imitation learning, and no fitness metric is utilized. It is possible that the approach implicitly discovers a fitness metric, and considers the suitability of multiple trajectories (i.e. performs planning), but it wasn’t made clear that that’s what’s happening. I would advise the authors to either avoid referring to their method as a planner, elucidate the connection between planning and their method, or to specify a different sense of the word “planning” to the one commonly assumed in AI literature. * The work is positioned as a graph-based planner that allows self-correcting actions, but excludes Tactical Rewind (Ke et. al., 2019, https://arxiv.org/pdf/1903.02547.pdf) from empirical comparison as it’s a decoding approach. Decoding in this context is a form of planning, using the model as a fitness metric. Given the way EGP is positioned as a planning approach, such relevant planning approaches should be compared to. Furthermore, Tactical Rewind includes similar ideas in that it iteratively builds a graph of the unseen environment, and uses it to select actions that allow for backtracking. I do think that EGP is a more general and potentially powerful approach and could conceivably learn the same backtracking behaviors that have been manually engineered in the Tactical Rewind approach, but perhaps that discussion should be included in the paper. * I would suggest that the authors formally describe the environment interface that their algorithm assumes is present. That is what is the full structure of observations at every timestep, and the action space, including the fact that graph connectivity is provided by the environment. Otherwise it seems that the paper assumes that the vision-and-language navigation problem is fundamentally graph-based, whereas in actual fact the navigation graphs are merely a feature of a popular dataset (Room-to-room). * The paper might benefit from an algorithm figure. * Nitpicking: Slightly incorrect to denote decision space by a set A, when the decision space is state-conditional. * Typos: Line 119: the->then Line 125: plannings->planning Line 193: instructions->word embeddings/representations Line 205: spurious comma ----- POST REBUTTAL COMMENTS ----- Thank you for providing the no-vision and no-language ablations! Could you please include them in the paper? The fact that performance without vision is relatively high shows that vision is not that crucial as a modality, however I don't think it takes away from the contribution of this paper, but is rather an artifact of the room to room dataset. To the contrary, the graph based approach seems particularly good at exploiting the graph structure, which is very interesting, and could have applications in other domains where graphical structure is more natural than in navigation. Thank you for addressing the comparisons to Ke et. al. and beam search. Yes, please add a more elaborate discussion. Boundaries between different policy classes can be subjective, and readers with different backgrounds will see the boundaries differently. For full context, the performance metrics of Ke et. al. should be included in the results table as well, but you can make it clear that EGP in comparison doesn't need backtracking rules, progress monitor, etc. It would make the paper more well rounded and more enjoyable to read for broader audiences.


Review 4

Summary and Contributions: This paper proposes a graph-based memory system and decision strategy for solving navigation tasks. The authors demonstrate their proposal on two popular vision-and-language navigation benchmarks. The main contributions of the paper are a strategy for reducing the decision space by considering only the leaf nodes of a navigation problem, and the use of pruning and pooling/unpooling techniques for scalability.

Strengths: The use of graphical architectures for memory and reasoning in navigation is a very natural and well-founded choice. Scalability is a problem for these approaches, and this paper proposes two significantly compelling techniques for mitigating scaling issues. Vision-and-language navigation in unseen environments is also a challenging and timely problem to tackle, so the paper is clearly relevant.

Weaknesses: In my view, this work has two significant flaws that prevent me from assigning it a higher score. 1. The graph-structured memory has no notion of loop closure. If the agent navigates around a loop back to the same place, how does it know? Is the data association between nodes given explicitly, or is each repeat visit simply treated as a new location? In any case, this requires discussion in the text, as loop closure is arguably the key challenge in online navigation. 2. The more significant flaw is potentially also the source of the paper's excellent results: the agent's decision space is limited to nodes on the frontier, and it can essentially "teleport" across the graph to get to those nodes immediately. This appears to be such an advantage from a learning and exploration point of view (by reducing the action space to only those that an oracle knows are promising) that it is not really surprising that the results are so much better than the baselines: the baselines do not have this advantage. At the bare minimum, this paper should either demonstrate ablation experiments where this frontier-only decision space is disabled in favor of an all-nodes decision space, or the baselines should be augmented with the same advantage; otherwise, the comparisons are not on equal footing.

Correctness: The claims and method appear to be correct. However, as stated above, the comparisons to baselines exhibit a crucial flaw in terms of the proposed agent being given privileged (and in the real world, unrealistic) knowledge of which nodes are frontier nodes and how to get there with perfect success. The baselines do not have this advantage, so some ablation or analysis of this strategy is strictly required.

Clarity: The paper is well-written and the ideas are good and clearly communicated.

Relation to Prior Work: Discussion of prior work is excellent, although additional effort should be made to illustrate the significant advantage that this technique has over previous work in terms of the privileged knowledge conveyed by the reduced decision space.

Reproducibility: Yes

Additional Feedback: This work is marginal in my view, but with the significant issue above addressed, I would consider this a definite accept and would argue in favor of it. ----- UPDATE ----- After considering the authors' feedback and discussing with the reviewers, I maintain my previous evaluation. Reasoning: While the authors did respond to my concerns, I am not convinced that I misunderstood anything about the authors' approach. The ability to unerringly select and navigate to leaf nodes in an environment is absolutely an "oracle" level capability, and it is not particularly surprising that an agent will perform better than competitors when those competitors are dealing with a more realistic decision space for an embodied agent: it is not clear that their methods would not perform equally well to the proposed approach if they were not simply using the same reduced decision space as that in this paper. In other words, it's not clear to me that the proposed architectural components are necessarily responsible for the improvements in performance, so much as the privileged information is. In addition, my concern about loop closure wasn't addressed: in the real world, perfect localization and mapping is extremely nontrivial. In addition, the reviewer discussion brought to light the issue of the Tactical Rewind paper of Ke et al. 2019. On the one hand, this provides precedent for reducing the decision space to the frontier, but on the other hand, it should absolutely be included in a comparison of the results.

[Author Response · NeurIPS 2020]

We thank the reviewers for their valuable comments. We will add the suggested clarifications to the paper and appendix.

**\*Reviewer 1** Thank you. We'll add more explanation about supervision, proxy graph, prior work, etc as suggested.

*Evaluation:* Internal travel during the graph jumps are also included in the path and path length (following the standard
in Beam search, all paths traveled are considered). Our model has lower PL due to design of the EGP (see Comp.).

*Supervising strategy:* (1) Starting with **empty set**, at the first step, the starting node and leaf nodes (STOP action +
connected neighbor nodes) are all added. The leaf nodes will contain the GT action. At next step, the GT action will be
kept as leaf node if not picked, or the agent visits the GT node, where the next GT action is connected. Hence, there
will be at least one GT action in the graph. (2) **Mismatch**: the potential supervision noise comes from the loss function
- the target actions computed with shortest path on new routes potentially contain nodes deviating from the expert path
(leading to mismatch), while ours doesn't. Agent indeed can explore the map, but the "correction" supervision from the
loss functions (computed via target node/action) should provide correct&matched signal. This intuition is also supported
by results in table 3 - *EGP-WithShortestPath*. (3) $\tau_t \cup (v)$ is created by connecting $\tau_t$ with jump path(L149-157).

*Proxy graph:* We will add vis&analysis. **Motivation**: (1) For every step, only a subset of nodes in the graph are relevant
for planning. Those nodes are selected and pooled to form the new graph, similar to how the attention mechanism is
used; (2) Some nodes that are far away and require long-distance communication in the original graph are bridged
together through proxy graphs. **More details:** (1) In R2R, the original graph leads to 51% SR, slightly lower than 52%
(proxy graph). In R4R, the long paths lead to large graphs and fail to fit in memory with decent batch size. (2) We use 6
nodes in proxy graphs.

*Comp. to beam search, Ke et al.:* **Model:** (1) Note that there is a distinction between generating a better policy distri-
bution vs. performing better maximum a posterior inference given a fixed joint distribution. For the latter, the algorithm
has no control over the distribution and is merely using extra cost (e.g. PL) to produce better MAP estimates. The
cost will be worse in longer trajectories. (2) EGP (the former) instead directly generates the distribution through using
feature-level information and is optimizable: the nodes in graphs communicates through high-dimensional messages to
propose solutions to minimize the loss func. Note that EGP still uses *greedy inference* in its own distribution form.
Although sharing some similarities, those methods belong to two difference classes. **Performance:** (1) Path length(PL)
is important in navigation. EGP performing well only using 64.7% PL comparing to Ke et al. and generalizable to
longer trajectories in R4R while maintaining normal PL. (2) Some more details: Ke et al. uses more information/designs:
progress monitor, speaker score, designs rules for backtrack and (optional) a reranker, which are not needed in EGP.

*"claim", 148, 163, tbl.3 mp:* (1) Thanks for pointing out. We meant "student forcing *with shortest-path supervision ...*".
(2) It's ranked by current policy. Our full model instead considers all actions. (3) Metric: nDTW [43]. (4) tbl.3 mp: $K_p$.

**\*Reviewer 2** Thank you for the suggestion. We will add qualitative results to illustrate the advantages of our model.

**\*Reviewer 3** Thank you. We'll add the discussion of prior work and of planning, and will address the writing comments.

*Two important ablations:* Compared to our full model SR (52%): (1) W/o language, EGP has 28% SR, showing that
EGP is best used with language; Monitor has 24%, indicating that EGP can exploit more long-range node information
(2) W/o vision, our model has 47% SR, which is 5% away from 52%, indicating EGP also relies on vision. Note that
EGP is a general graph module that contributes to better exploiting structured knowledge(angle, vis). The vision issue
might be inherited from base agent[14] and problem settings(sparse topological connections, pre-computed features).

*Fitness metric&position:* The graph module in EGP is implicitly maximizing the nDTW score through proposing nodes
that align with the language, this is implicitly learned through loss func using the target node generated by our strategy
(L163-164). We view [43] and beam search as a better MAP inference tech compared to using differentiable planner to
generate a better distribution (see **R1** Comp.). But indeed it can also be treated as planning. We'll add a full discussion.

**\*Reviewer 4** Thank you. We will clarify the potential misunderstandings in the paper.

*Oracle advantages:* EGP does not use an oracle to select nodes and the action space is not reduced. (1) EGP chooses
either to visit a new location or stop at an internal node. The "stop at this internal node" action is treated as a leaf node
for consistency of representation (will update fig.1 to clarify this). So the agent is already dealing with all nodes. (2)
The baselines are exposed to the same amount of information but don't have a module to utilize it.

*Success at teleport:* The environment (not our model) defines the primitive action that agents can successfully perform
(e.g. "Pickup", make relative movement (+4, -1) in other envs). Our agent jumping to another location is analogous to
making consecutive moves ((+4, -2), (0, +1), ...) w/o mistakes in other envs. We agree that in reality this will all need
more considerations due to noisy locomotion, but our model can potentially still serve as a high-level planning module.

*Loop closure:* The agent's hidden state encoded in memory contains visited nodes and indicates the revisit case. Our
graph network module will then also identify the revisit of internal nodes based on memory and node feature.

[Meta-Review · NeurIPS 2020]

This paper addresses the problem of vision-and-language navigation from raw visual input and language instructions in a photorealistic indoor environment (Room-to-Room) by iteratively building a high-level graph representation and then goal-driven planning using Graph Neural Networks. Instead of planning on the full graph, the model predicts actions over the fringe nodes of that graph (i.e., jumps through the graph using shortest path) and it also predicts and plans on a sparser proxy graph representation (these are novel ideas). It is trained using imitation learning. After discussion and authors' rebuttal, the reviewers' scores are (6, 7, 7, 6). While many of the reviewers' concerns are addressed, the main remaining concerns are a missing comparison to graph search methods (specifically: "Tactical Rewind: Self-Correction via Backtracking in Vision-and-Language Navigation"), confusion about the word planning in an imitation learning setting, discussions about how loop closure is performed, acknowledging the competitive advantage of knowing which nodes of the graph are frontier nodes, and scarce information about how to reproduce the work. Based on these comments, I recommend acceptance as spotlight or poster, and expect the authors to hold on their promises of including algorithmic details in an expanded appendix, ablations, and also add a comparison with Tactical Rewind.